# A high-strength silicide phase in a stainless steel alloy designed for wear-resistant applications

D. Bowden [1], Y. Krysiak[2], L. Palatinus [3], D. Tsivoulas[1,4], S. Plana-Ruiz[2,5], E. Sarakinou[1,6], U. Kolb [2], D. Stewart[7] & M. Preuss [1]

Hardfacing alloys provide strong, wear-resistant and corrosion-resistant coatings for extreme environments such as those within nuclear reactors. Here, we report an ultra-high-strength Fe–Cr–Ni silicide phase, named π-ferrosilicide, within a hardfacing Fe-based alloy. Electron diffraction tomography has allowed the determination of the atomic structure of this phase. Nanohardness testing indicates that the π-ferrosilicide phase is up to 2.5 times harder than the surrounding austenite and ferrite phases. The compressive strength of the π-ferrosilicide phase is exceptionally high and does not yield despite loading in excess of 1.6 GPa. Such a high-strength silicide phase could not only provide a new type of strong, wear-resistant and corrosion-resistant Fe-based coating, replacing more costly and hazardous Co-based alloys for nuclear applications, but also lead to the development of a new class of high-performance silicide-strengthened stainless steels, no longer reliant on carbon for strengthening.

[1] School of Materials, The University of Manchester, Oxford Road, Manchester M13 9PL, UK. [2] Institute of Inorganic Chemistry and Analytical Chemistry, Johannes Gutenberg University Mainz, Jakob-Welder-Weg 11, 55099 Mainz, Germany. [3] Academy of Sciences of the Czech Republic, Institute of Physics, Na Slovance 2, 18040 Praha 8, Czech Republic. [4] Wood plc, 601 Faraday Street, Birchwood Park, Warrington WA3 6GN, UK. [5] LENS, MIND/IN2UB, Electronics and Biomedical Engineering, Faculty of Physics, University of Barcelona, Martí i Franquès, 1-11, 08028 Barcelona, Catalonia, Spain. [6] Interface Analysis Centre, University of Bristol, Bristol BS8 1TL, UK. [7] Rolls-Royce plc, Derby DE24 8BJ, UK. Correspondence and requests for materials should be addressed to M.P. (email: michael.preuss@manchester.ac.uk)

Increasing energy supply demands, along with reductions in the availability and use of conventional fossil fuels, has led to significant research and investment into future nuclear power solutions. A drive by the industry to reduce contamination and increase safety while remaining cost-effective has been a prime motivator when considering future nuclear plant builds. Traditionally, Co-based hardfacing materials, such as Stellite alloys, have been used extensively in the nuclear sector. Co-based hardfacing materials are wear-resistant, strong and corrosion-resistant materials[1] that can be applied as a coating to a structural component such as a pump or valve. However, Co is both costly and hazardous when irradiated, forming $^{60}$Co which contributes the largest dose of radiation exposure to nuclear plant personnel[2]. Hence, cobalt-free hardfacing materials are being developed as replacement alloys for wear-resistant applications within the nuclear power generation sector[2–4]. RR2450 is a highly alloyed, complex Fe-based alloy that has been designed to replace the Stellite alloy family as a Co-free alternative. RR2450 was developed by Rolls-Royce as a derivative of the Fe-based stainless steel alloy Tristelle 5183[5].

Hardfacing alloys rely on several mechanisms to produce a strengthening effect (and therefore enhanced wear resistance). The first mechanism is by way of work hardening, typically through a strain-induced transformation such as: austenite → ε-martensite → α'-martensite. This strain-induced transformation relies on a low stacking fault energy (SFE), which is believed to lead to the evolution of the high-wear resistance exhibited by Co-based alloys at both low and high temperature wear testing[3]. A low SFE results in a face-centred cubic matrix with a greater propensity for generating widely spaced, numerous stacking faults, which act to impede dislocation cross-slip. Second, a large fraction of hard secondary phases is often generated[6]. These usually consist of carbides and nitrides, which are both inherently strong and potentially fine enough to pin matrix dislocations. However, a large fraction of Cr-carbides, which typically evolve in high-carbon stainless steels, can lead to localised corrosion through matrix Cr depletion, called sensitisation[7]. Solution strengthening of the alloy matrix is also an effective way to increase the alloy wear resistance, which through atomic size mismatch, acts to introduce strain fields within the matrix, increasing the energy required to enable dislocation glide through these regions. Finally, control of the microstructure morphology serves as a method of strengthening and it is known that finer grain sizes enhance alloy strength as described by the Hall–Petch relationship. Thus far, the candidate Co-free replacements for hardfacing applications have focussed on generating a low SFE matrix and high fraction of carbide phases[8], the formation of which must also avoid leaving the alloy matrix in a sensitised condition. These attempts have so far been unsuccessful at replicating the combined high-temperature wear and corrosion resistance of the Co-based alloys[9].

Understanding the microstructure of an alloy such as RR2450 is the only means of ensuring that the alloy can be reproduced in a stable and predictable manner. Additionally, an understanding of the fraction of hard versus soft phases, potentially also related to the type of manufacturing route, can provide an early insight into the mechanical properties of an alloy. RR2450 is initially produced in gas-atomised powder form, which is placed into a pre-formed container to undergo hot isostatic pressing (HIP). During HIP, high temperatures (up to 2200 °C) and high pressures (up to 300 MPa) are applied to a container filled with the powder[10,11]. From this, a consolidated component with a near-net-shape (NNS) finish can be produced. The shapes produced through the use of HIP vary in complexity and size and HIP can be utilised to bond several components together[11]. In this case, the HIP consolidated RR2450 hardfacing coating would be subsequently HIP bonded to a structural component such as a valve seat. HIP is the preferred manufacturing route for such components owing to the NNS end product, homogenous microstructure and mitigation of issues arising from more traditional welding application techniques which can lead to thermally induced cracking and variation in coating quality through poor process control and heavy reliance on operator skill[12].

In this study, we report the existence of a silicide phase which forms within the austenite/ferrite matrix in the RR2450 alloy. The structure is determined by using electron diffraction tomography (EDT) coupled with the software package Automated Diffraction Tomography 3D (ADT3D)[13–17], and accurately refined by the dynamical refinement technique[18–20] using the software packages PETS[21] and JANA2006[22]. We discuss the use of both kinematical and dynamical EDT refinements to establish the silicide phase crystallography, and how further research around silicide-based strengthening has the potential of changing our strategy when developing future high-strength steels.

## Results

**Identification of the silicide phase**. Initially, the phase fractions of the HIP consolidated RR2450 alloy were measured using laboratory X-ray diffraction (XRD). The compositions of both the RR2450 alloy and its parent Tristelle 5183 are shown in Supplementary Table 1. The total phase fractions evolved in the HIP consolidated RR2450 alloy are shown in Supplementary Table 2. RR2450 is designed to produce a duplex austenite (γ) and ferrite (δ) microstructure[5]. Since RR2450 contains 1.8 wt% C (Supplementary Table 1), a high fraction of carbides of the form (Nb,Ti) CN and $M_7C_3$, (where M denotes a mixture of Cr and Fe) also evolve within the microstructure. A crucial result for the RR2450 alloy is that it also contains up to 23% of a silicide phase (Supplementary Table 2) resulting in the formation of a triplex δ/γ/silicide matrix. A typical XRD powder pattern collected for the RR2450 alloy is shown in Fig. 1.

The silicide phase was initially identified from electron backscattered diffraction (EBSD) data collected using a scanning electron microscope (SEM). The light blue regions on the EBSD phase map shown in Fig. 2a correspond to silicide phase grains present in the RR2450 alloy (Fig. 2b). The silicide phase regions are shown in the energy dispersive X-ray spectroscopy (EDS) maps collected in the SEM (Fig. 2c-e). We found the regions of silicide phase identified from the EBSD map to have high concentrations of Fe, Cr, Si and Ni. EDS point analysis revealed an average composition of 53Fe–22Cr–15Ni–10Si (wt%) / 48Fe–21Cr–13Ni–18Si (at%) for the silicide phase (Supplementary Fig. 1), which was also confirmed by scanning transmission electron microscopy (STEM)-based EDS analysis.

For more reliable quantitative elemental analysis, wavelength dispersive spectroscopy (WDS) was carried out using an electron probe microanalyser (EPMA). The EPMA-WDS results for the silicide phase composition, also shown in Supplementary Fig. 1, were again in very good agreement with the SEM and STEM-EDS data. However, unlike EDS, the WDS data can also provide quantitative evidence for light elements such as carbon by accounting for compositional difference in stoichiometry. Consequently, the EPMA-WDS data revealed the presence of up to 1.2(4) wt% carbon within the silicide phase (Supplementary Fig. 1). Errors quoted within this article are the standard uncertainty of the mean.

**The structure of the π-ferrosilicide phase**. With the phase chemistry established, we moved on to determine the crystallography of the silicide phase using the EDT technique[14–16]. This process is outlined in schematic form in Supplementary Fig. 2.

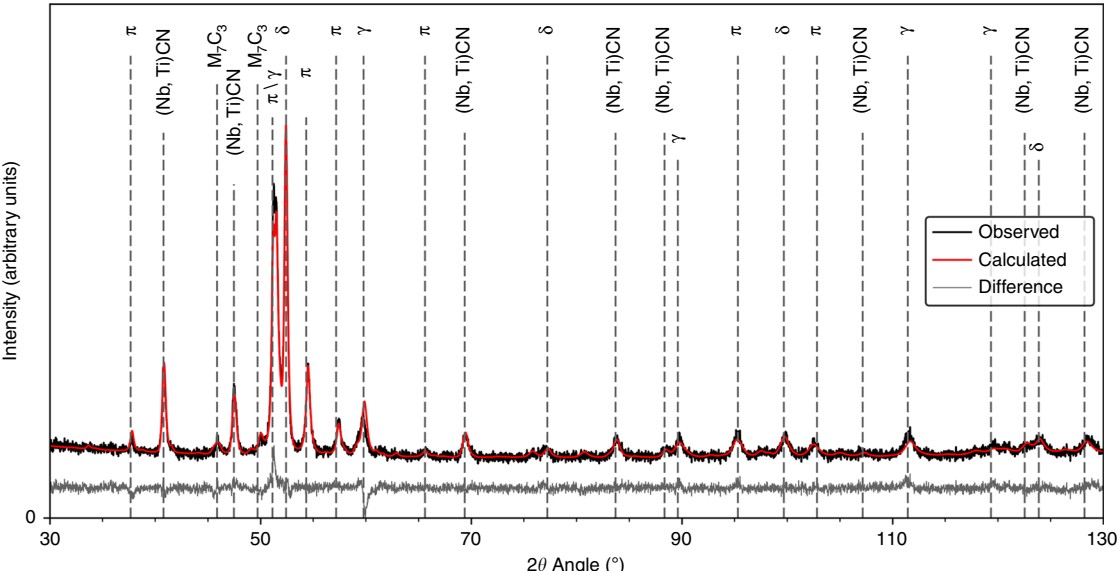

**Fig. 1** Powder XRD pattern for the RR2450 alloy. Measured using a Co source ($\lambda = 1.79$ Å). Major peaks have been marked according to their corresponding phase: austenite ($\gamma$), $\delta$-ferrite ($\delta$), silicide ($\pi$), $M_7C_3$ and (Nb,Ti)CN. Reported goodness of fit (GOF) = 1.35

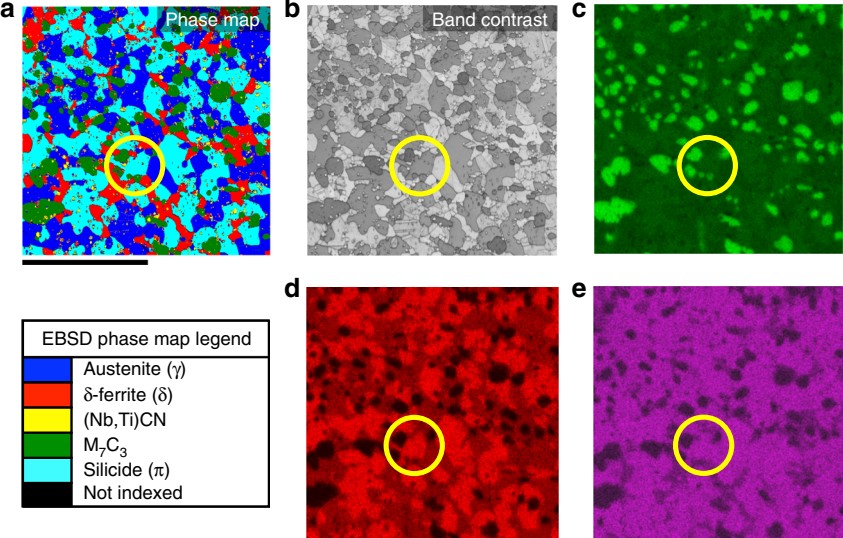

**Fig. 2** Elemental analysis data for a typical region of the RR2450 alloy. Scale bar of 25 μm is shown. A silicide phase grain is circled for clarity on each map. **a** EBSD phase map; **b** band contrast image; **c** Cr SEM-EDS map; **d** Si SEM-EDS map; **e** Ni SEM-EDS map

The data were processed by the software packages PETS[21] and JANA2006[22]. Initially a kinematical refinement was performed. From this, we found that the silicide phase crystallography matched that of a primitive cubic setting; space group $P2_13$ and corresponds to an $Au_4Al$ structure type[23]. The silicide phase lattice parameter was determined from Rietveld refinement of the RR2450 XRD powder pattern to be 6.1908(1) Å. The refinement yielded a crystallographic residual value R1 = 20.3%.

In addition, the kinematical refinement of the EDT data appeared to suggest the presence of extra maximum in the potential map, located in the tetrahedral voids, although the height of the maximum barely exceeded the level of $3\sigma$ of the difference potential map (circled in Fig. 3a). These sites of additional potential were initially believed to indicate the presence of interstitial carbon. However, the kinematical approximation often produces artefacts which can be mistaken as genuine features.

Dynamical refinement[18–20] was therefore carried out on the same data set. The dynamical refinement provides better structure models and it should eliminate artefacts. It proceeded smoothly and converged at R1 = 7.03%. Further information about the dynamical refinement is summarised in Supplementary Table 3. The refined structure was in agreement with the silicide phase crystallography established by using the kinematical approach. The dramatic improvement of the quality of fit is reflected also in the reduction of the noise level in the difference potential map by a factor of about 10 (Fig. 3b). The difference potential map obtained by the dynamical refinement does not contain any significant maxima in the tetrahedral voids and hence does not confirm the presence of the additional atomic site. It can thus be concluded that the additional maximum is an artefact introduced by the use of the kinematical approximation. Within the sensitivity of the electron diffraction data, no well-defined

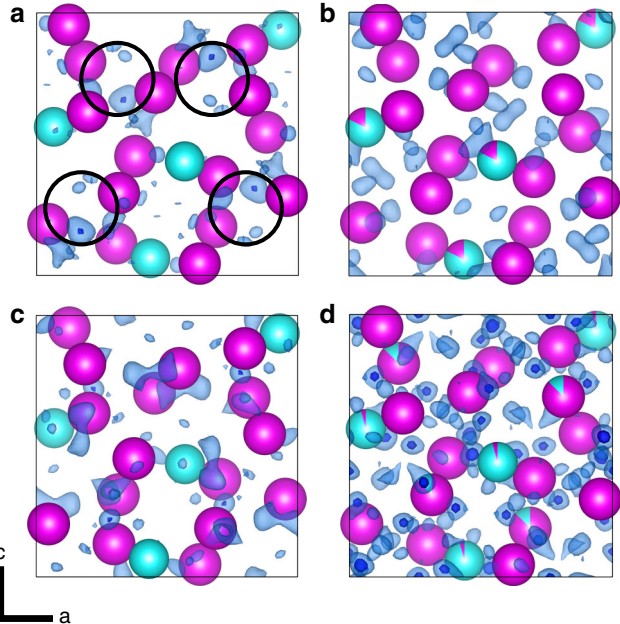

**Fig. 3** Results of ab initio silicide phase refinements plotted using VESTA V3.4[43]. Fe, Cr and Ni atomic sites are interchangeable and shown in magenta. Si atomic sites are shown in turquoise. Residual Coulomb potential maps ($\Delta V$) of the RR2450 π-ferrosilicide phase and carbon-deficient π-ferrosilicide phase solved in the $P2_13$ space group setting using the output of the refinements from JANA2006[22] are shown. Transparent light-blue residuals are plotted using a $2\sigma(\Delta V)$ threshold ($\sigma$ is the standard deviation of map values) and opaque dark-blue residuals are plotted using a $3\sigma(\Delta V)$ threshold. Maps shown are of: **a** π-ferrosilicide after kinematical refinement, $\sigma(\Delta V) = 1.09$ e/Å. Regions initially believed to correspond to a 25% carbon occupancy are circled; **b** π-ferrosilicide after dynamical refinement, $\sigma(\Delta V) = 0.093$ e/Å; **c** carbon-deficient π-ferrosilicide after kinematical refinement, $\sigma(\Delta V) = 1.41$ e/Å, and; **d** carbon-deficient π-ferrosilicide after dynamical refinement, $\sigma(\Delta V) = 0.054$ e/Å. Note that the noise levels in the dynamical difference potential maps, quantified by the values of $\sigma(\Delta V)$, are more than ten times lower than in the kinematical difference potential maps

additional position for a carbon atom can be identified in the structure.

This analysis is a further indication that the results of the kinematical treatment of EDT data need to be interpreted with great caution. However, there is a strong indication that the π-ferrosilicide phase in RR2450 does contain a small amount of carbon, both from the chemical analysis using WDS (Supplementary Fig. 1) and by the results of the decomposition experiments at high temperature (see below).

The dynamically refined structure of the π-ferrosilicide phase is shown in Fig. 4. Refined coordinates of the atomic sites are listed in Supplementary Table 4 and calculated interatomic distances are shown in Supplementary Table 5. Interestingly, the refinement showed that approximately 18% of the Si position is occupied by a metal atom (Fe, Cr and/or Ni). Such non-stoichiometry is, however, still within the limits of the compositional ranges for this phase, as observed by the scatter in Supplementary Fig. 1.

It is therefore concluded that the silicide phase present within the RR2450 alloy is a close match to the τ-phase identified in the Fe–Ni–Si system, which is isostructural to the π-phase in Cr-based alloys, possessing a space group setting of $P2_13$[24–26]. Therefore, it is proposed that the silicide phase identified in the

RR2450 alloy, be named π-ferrosilicide, given that it contains both Fe and Cr, as well as Ni and Si.

For the sake of comparison, we also analysed a sample with the π-ferrosilicide phase composition but with carbon deliberately excluded from the stoichiometry (henceforth referred to as carbon-deficient π-ferrosilicide). The sample was extracted from a cast ingot of carbon-deficient silicide, with a composition of 53Fe–22Cr–15Ni–10Si (wt%). Laboratory XRD confirmed that the carbon-deficient ingot contained 83% π-ferrosilicide phase with the remainder being δ-ferrite (Supplementary Table 2). As before, we performed lattice reconstruction of a π-ferrosilicide phase grain from this ingot using EDT with both kinematical and dynamical refinements. The kinematical refinement performed in JANA2006[22] confirmed the Au₄Al structure type (Fig. 3c). The results of the dynamical refinement are reported in Supplementary Table 3 along with the refined atomic positions shown in Supplementary Table 6 and calculated interatomic distances in Supplementary Table 7. The difference map from the dynamical refinement did not show any significant features indicating additional atomic positions (Fig. 3d). The refined structures of carbon-containing and carbon-deficient π-ferrosilicide are therefore very similar.

However, we do observe a decrease in cell size in the carbon-deficient π-ferrosilicide to 6.1669(2) Å, as observed from Rietveld refinement of the XRD data. The difference in lattice parameters equates to a 2.739 Å³ (1.168%) volume expansion of the carbon-containing π-ferrosilicide compared to the carbon-deficient π-ferrosilicide. This observation strengthens the argument of carbon solubility, which is not detectable by EDT, whereby carbon is located interstitially within the π-ferrosilicide phase.

**Hardness testing**. Given the intended hardfacing application of the RR2450 alloy, it was useful to compare the hardness of the carbon-containing π-ferrosilicide and carbon-deficient π-ferrosilicide phase to that of other phases in the RR2450 and silicide ingot alloys. The average macrohardness of different batches of the RR2450 alloy was found to be 557(21) HV (using a 50 kg load). Arrays of 100 nanohardness indents were made into the surface of both RR2450 and the carbon-deficient silicide phase ingot to test individual phase hardness values (Supplementary Fig. 3). The average nanohardness measurements for each phase within the RR2450 alloy and the carbon-deficient silicide phase ingot are shown in Fig. 5a. Indents that were made at a phase or grain boundary were excluded from this analysis in order to avoid boundary effects affecting the recorded average.

Only π-ferrosilicide phase hardness values were obtained from the carbon-deficient ingot as many of the indents fell too close to δ-ferrite/ π-ferrosilicide phase grain boundaries, preventing an accurate hardness reading of the δ-ferrite phase islands.

The highest hardness values in the RR2450 alloy were unsurprisingly recorded at carbide indent locations. A hardness of 3136(75) HV was found for (Nb,Ti)(C,N) and 2642(42) HV for $M_7C_3$. However, we were extremely surprised to see an average hardness of 1680(74) HV recorded for the π-ferrosilicide phase in RR2450, between 2 to 2.5 times harder than the δ-ferrite and austenite phases respectively (Fig. 5a). The hardness of the carbon-deficient π-ferrosilicide was comparable to the carbon-containing π-ferrosilicide, with an average hardness of 1549(21) HV. The comparable hardness of the carbon-rich/carbon-deficient π-ferrosilicide phases is an indication that the presence of carbon is not a pre-requisite for enhanced strength in this particular phase.

It is noted that the nanohardness measurements of the silicide phases contain quite significant scatter which is attributed to two possible reasons. First, the effects of crystallography introduce

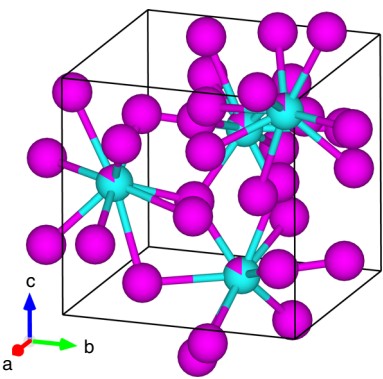

**Fig. 4** The RR2450 π-ferrosilicide phase structural solution after a dynamical refinement. Fe, Cr and Ni atomic sites are interchangeable and shown in magenta. Si atomic sites are shown in turquoise. Up to 18% occupancy of Fe, Cr or Ni on the Si atomic sites is indicated in the schematic, which is plotted in VESTA V3.4[43]

variations in the recorded hardness, given the existence of so called hard and soft orientations depending on the presence of readily available slip-systems in the loading direction. Additionally, due to the complex nature of this microstructure (Fig. 2), it is probable that sub-surface features such as carbides have inadvertently been sampled, giving rise to artificial hardness increases. These effects would explain the observed scatter in hardness value for the π-ferrosilicide and carbon-deficient silicide phases (Fig. 5a).

**Compressive strength**. To further assess the mechanical properties of the RR2450 alloy and the π-ferrosilicide phase, neutron diffraction measurements were carried out on samples of the RR2450 alloy undergoing in-situ compressive loading up to a maximum load of 100 kN. The setup used for this experiment is shown in schematic form in Supplementary Fig. 4. Compressive loading represents the loading case which is most likely to be experienced by such hardfacing alloys in service within valve and pump assemblies. Compressive elastic strains evolved in the axial direction within the austenite, δ-ferrite, (Nb,Ti)CN and π-ferrosilicide phases, versus compressive stress are shown in Fig. 5b. It should be noted that data for the $M_7C_3$ phase has not been plotted due to the anisotropic nature of its orthorhombic unit cell lattice parameters. By observing the deviation of elastic microstrain from linearity, as compressive stress increases, it can be ascertained when a phase begins to yield (i.e. transition from an elastic to plastic mode of deformation)[27]. In the case of the RR2450 alloy, austenite is shown to yield at 410 MPa and δ-ferrite at 853 MPa (Fig. 5b).

However the π-ferrosilicide, in a similar fashion to (Nb,Ti)CN, does not yield, despite loading to a compressive stress in excess of 1.6 GPa. In the case of these hard phases, load is partitioned to them as the surrounding, weaker phases yield and can no longer accommodate a further increase in stress. It is therefore apparent that the π-ferrosilicide does not yield, or even appears to undergo brittle failure during compressive loading to extremely high stresses, further testament of the exciting possibility of utilising silicide strengthened steels in future high-strength applications.

**Phase formation and stability**. The phase-change behaviour of the RR2450 alloy during a HIP cycle was assessed by collecting synchrotron XRD powder patterns during in-situ heating of the gas-atomised alloy powder. A typical HIP cycle profile, used to consolidate the RR2450 alloy is shown schematically in

Supplementary Fig. 5. The results from this study, shown in Fig. 6, reveal that the π-ferrosilicide decomposes to form δ-ferrite and $M_7C_3$ carbides once the HIP cycle temperature exceeds 520 °C during the heating portion of the cycle. The evolution of $M_7C_3$ carbides during π-ferrosilicide decomposition further supports the suggestion of carbon solubility within the π-ferrosilicide. When a temperature of 920 °C is reached, the π-ferrosilicide is fully decomposed and remains absent throughout the dwell portion of the HIP cycle, before precipitating below 920 °C during cooling (Fig. 6). These findings are in agreement with those of the τ-phase, described in the literature[24,26]. The precipitation of π-ferrosilicide during the cooling part of the HIP cycle occurs at the expense of δ-ferrite.

The strong dependence of π-ferrosilicide formation on the presence of δ-ferrite is further reinforced by the chemical analysis and phase fraction quantification by laboratory XRD across a range of RR2450 powder compositions. Four batches of RR2450 were atomised, providing variations in alloy stoichiometry, allowing a comparison against the austenitic parent alloy Tristelle 5183, which typically contains a low π-ferrosilicide phase fraction as shown in Supplementary Table 2. There is a clear trend of increasing π-ferrosilicide phase fraction, at the expense of austenite, as shown in Fig. 7a. Restricting the formation of austenite allows ferrite to evolve, promoting the formation of greater quantities of π-ferrosilicide (Fig. 7b). The relationship between ferrite and π-ferrosilicide is further enhanced by a strong orientation dependence of the two phases. Supplementary Figure 6 shows how two orientation relationships (OR) between ferrite and π-ferrosilicide could be identified from EBSD data collected from the cast carbon-deficient π-ferrosilicide ingot. These OR were established to be $\{100\}_\delta || \{110\}_\pi$, $\langle 010 \rangle_\delta || \langle \overline{1}13 \rangle_\pi$ and $\{100\}_\delta || \{120\}_\pi$, $\langle 010 \rangle_\delta || \langle \overline{2}10 \rangle_\pi$. The existence of these ORs between ferrite and π-ferrosilicide will promote a greater interfacial strength between these phases due to their coherency. This is in contrast to carbide phases which can lack this coherency with the matrix, potentially leading to the pull-out of such carbides due to the reduced strength at the interface[28].

As the RR2450 alloy Ni content is increased, more austenite is stabilised as expected (Fig. 7c). Since greater levels of austenite formation act to restrict ferrite formation, this means increased Ni content has the effect of limiting the extent to which π-ferrosilicide is able to evolve (Fig. 7d). Conversely, increasing alloy Si content has the effect of stabilising greater quantities of the π-ferrosilicide phase as shown in Fig. 7e. Finally, Cr content was found to have little correlation with the amount of π-ferrosilicide evolved (Fig. 7f), which is understandable since π-ferrosilicide is isostructural to the Fe–Ni–Si τ-phase and therefore Cr is acting only to substitute for Fe and Ni sites in the lattice. It is clear that a competition exists between the Ni and Si contents of these alloys, where increases in Ni content lead to increased austenite fractions at the expense of π-ferrosilicide. On the other hand, increased Si concentrations, at the expense of Ni, allow the triplex microstructure to evolve and promotes the formation of the π-ferrosilicide phase.

## Discussion

This study has demonstrated the use of EDT to determine the crystal structure of a newly-identified high-strength phase in steel, named π-ferrosilicide. We have successfully identified and indexed the lattice parameters and atomic positions of this phase, which corresponds to the $P2_13$ space group. Using EDT to study microscopic-isolated crystals circumvents the difficulties arising from analysis of XRD powder patterns in complex alloys containing intermetallics, such as the RR2450 alloy. The sensitivity of the EDT technique to identify variation of Coulomb potential

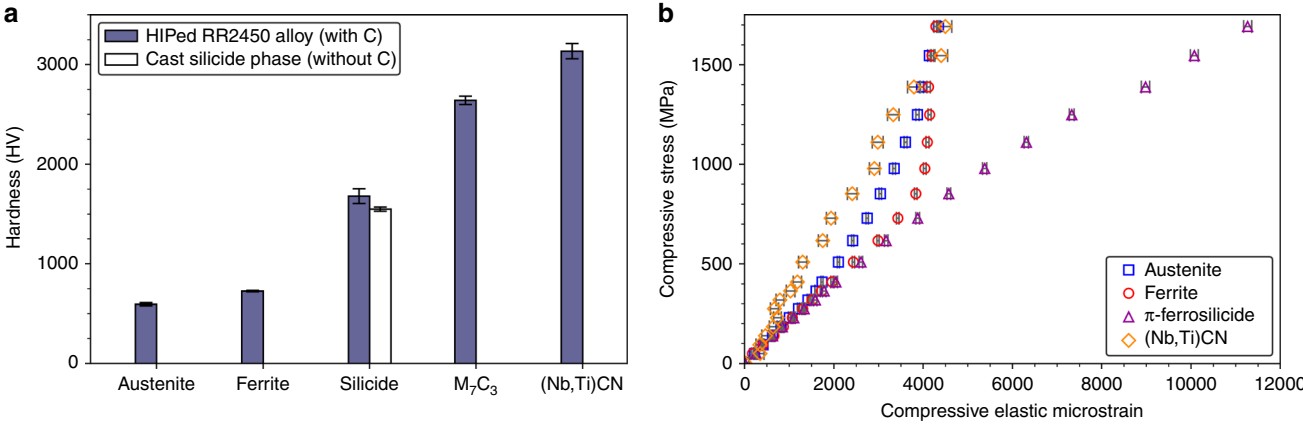

**Fig. 5** Mechanical test data for the RR2450 alloy. **a** The average Vickers nanohardness of the phases present in the RR2450 alloy and carbon-deficient silicide phase ingot. **b** The compressive elastic microstrain of the austenite, ferrite, π-ferrosilicide and (Nb,Ti)CN phases within the RR2450 alloy measured using neutron diffraction alongside in situ compression testing

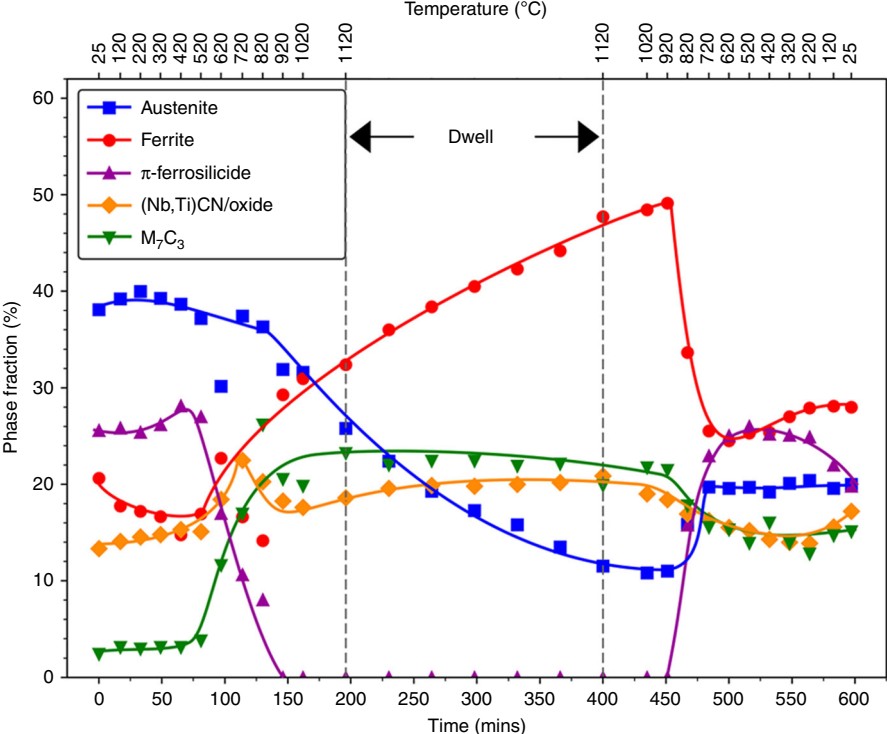

**Fig. 6** Time and temperature versus phase fraction during a simulated HIP cycle on the RR2450 alloy powder. Phase analysis was carried out on the synchrotron XRD powder patterns collected during the in-situ simulated HIP cycle. The π-ferrosilicide phase is shown to be fully decomposed above temperatures of 920 °C and precipitates $M_7C_3$ as it decomposes

within a unit cell lends its use favourably to establishing the presence of light elements, although as highlighted in this study, caution must be applied when assessing purely kinematical data. In this study, a dynamical refinement has showed that an extra carbon atomic site was initially incorrectly identified from the kinematically refined data set.

However, the elemental analysis, particularly with the support of more sensitive quantitative approaches such as EPMA-WDS, suggests the presence of 1.2(4) wt% carbon within the π-ferrosilicide phase. The increase in unit cell volume of the carbon-containing π-ferrosilicide, compared to the carbon-deficient variant, supports this result. An in situ heat treatment of gas-atomised RR2450 powder showed that π-ferrosilicide fully

decomposes at 920 °C and that this decomposition occurs in parallel with the evolution of $M_7C_3$ precipitates. These observations further support the conclusion of carbon solubility within the π-ferrosilicide phase, likely occurring by way of an interstitial mechanism. However, the determination of specific carbon sites to the level of ~1 wt% is beyond the capacity of EDT, even if coupled with the dynamical refinement, and it would also be extremely difficult, if not impossible to detect, by other types of diffraction techniques.

Of course, observing silicide phases within steel is not new, as reported previously[29]. However, the composition of the π-ferrosilicide is unique and more importantly, the discovery of this phase and its properties raises some interesting and exciting

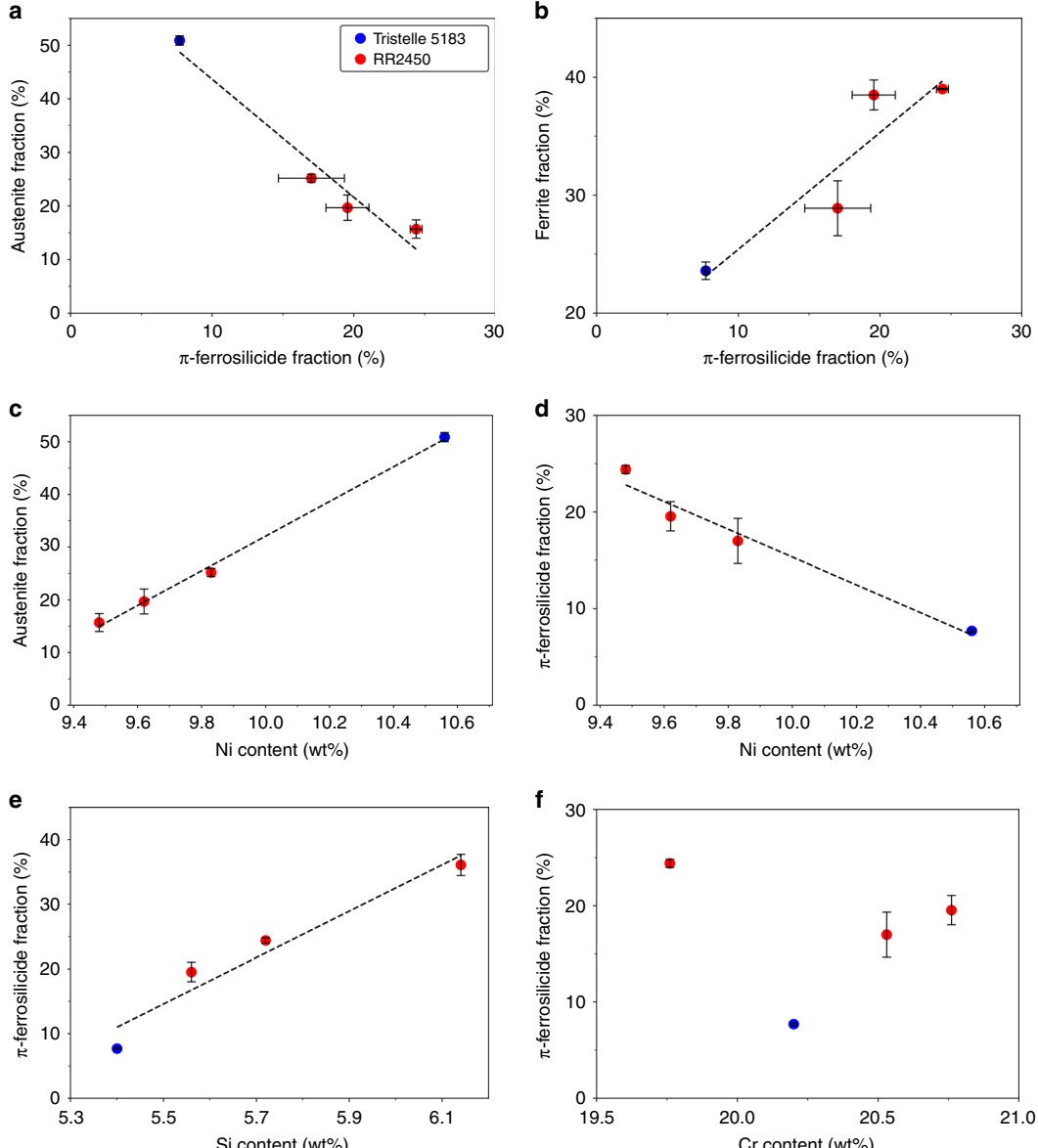

**Fig. 7** The variation of stoichiometry and the effect of phase stability for a range of RR2450 alloys and the original parent Tristelle 5183 alloy. **a** Relationship between π-ferrosilicide and austenite fraction; **b** variation of π-ferrosilicide fraction and the effect on ferrite fraction evolved; **c** effect of Ni content on austenite stability; **d** effect of Ni content on π-ferrosilicide stability; **e** variation of Si content and resulting π-ferrosilicide fraction; **f** effect of Cr content on π-ferrosilicide fraction evolved, note the lack of clear trend in this data set

prospects in the production of high-strength steels. We have shown that we can produce ingots composed almost entirely of this phase. This means that we can manufacture a readily available, extremely strong, corrosion-resistant steel alloy with relative ease.

Hardfacing alloys such as Stellite 6 are usually designed around carbide formation[6]. By designing stainless steel alloys around the use of silicide phases instead, carbon levels could be reduced, if not altogether eliminated, since it would be no longer necessary to rely on carbides to provide alloy strengthening. If Cr-carbide precipitation is restricted, we can reduce the risk of matrix sensitisation. With small additions of carbon, these alloys could be tailored to provide a higher strength than their carbon-deficient counterparts by introducing controlled, low volume fractions of carbides. As a result of this study, we believe that silicide-based and other unconventional stainless steels present viable

candidates for high-strength, corrosion-resistant alloys for use in a wide range of possible future applications, not just limited to replacing traditional hardfacing alloys.

## Methods

**Alloy production**. Samples of the stainless steel RR2450 alloy were initially nitrogen gas atomised by Sandvik Osprey, UK, who operate their atomisers using proprietary parameters, which cannot be elaborated on within this article. Cylindrical containers of 40 mm diameter and 100 mm height are fabricated and filled with the gas-atomised powder. The canisters are then vacuum-sealed ready for the HIP. The exact parameters used during HIP consolidation of the RR2450 are commercially sensitive parameters used by Rolls-Royce which cannot be outlined in detail but they are in a similar range compared to HIP consolidation of austenitic steel.

The carbon-deficient silicide ingot was cast using a custom made, Arcast laboratory vacuum arc furnace at Imperial College London. Starting materials of 99.9% purity were used to produce an ingot of composition 53Fe–22Cr–15Ni–10Si (wt%). A total material weight of 200 g was cast.

**Laboratory X-ray diffraction**. From the HIP canister, it was possible to cut a slice radially along the centre, suitable for a bulk phase analysis using X-ray diffraction (XRD). This was carried out using a Philips X'Pert diffractometer operating with a Co ($\lambda = 1.79$ Å) source and fitted with a graphite monochromator. The data collected was subsequently analysed with TOPAS version 4.2[30] using a Rietveld refinement with the pseudo-Voight Thompson–Cox–Hastings peak shape applied. A fully fitted XRD powder pattern for the RR2450 alloy is shown in Fig. 1.

**Electron microscopy analysis**. A smaller sample of RR2450 suitable for a SEM was cut from the HIP canister. This sample was ground down to 4000 grit using silicon carbide paper, and polished down to 0.25 µm using a diamond spray suspension. The final polishing step was carried out on a Buehler VibroMet2 vibratory polisher using a colloidal silica suspension with particle sizes between 22 and 28 nm. The sample was left to vibration polish for 3 h. Using the vibratory polisher helps to reduce the distance that hard particles protrude from the comparatively soft matrix in the sample. A sample of the carbon-deficient silicide phase ingot was also prepared in the same way.

The π-ferrosilicide phase was identified by EBSD using a Quanta 650 field emission gun (FEG) SEM operating at an accelerating voltage of 20 kV, which produced intense Kikuchi bands suitable for more reliable pattern indexing. A step size of 0.12 µm was chosen across an area of ~60×60 µm. This allowed sufficient indexing of larger matrix grains within the RR2450 alloy, without requiring unnecessarily long EBSD mapping durations. Due to the larger grain sizes within the cast carbon-deficient silicide phase ingot, a larger step size of 0.25 µm was used over an area of approximately 110×110 µm. Kikuchi patterns were collected using an Oxford Instruments Nordlys detector running using the Aztec analysis suite.

Once the EBSD map was complete and while the sample was still at the same position in the SEM, we could carry out EDS allowing us to determine the composition of the π-ferrosilicide phase within the RR2450 alloy. Regions of π-ferrosilicide could be analysed with EDS by using the previously collected EBSD phase map to locate specific π-ferrosilicide grains, while avoiding beam interaction with carbides at the sample surface, visible from a backscattered electron (BSE) image. The SEM-EDS data were collected with an Oxford Instruments silicon drift detector (SDD) X-Max$^N$ with 50 mm$^2$ area.

EDS data was also collected on a lamella of the RR2450 alloy, using a transmission electron microscope (TEM), using an FEI Tecnai T20 TEM, operating at 200 kV in scanning (S)TEM mode. An Oxford Instruments X-Max$^N$ 80 T SDD detector operated via the Aztec analysis software package was used for collection of this data. Automatic elemental identification was enabled for both SEM and STEM-EDS analysis, but readings for carbon were excluded since such values are unreliable from SEM-EDS analysis due to hydrocarbons present on the sample surface, producing a strong carbon k-edge peak in the collected EDS spectra.

WDS was applied to quantify the amount of carbon present in the π-ferrosilicide phase, although such results should also be interpreted with caution, since surface hydrocarbons can also produce false-positive carbon readings as with EDS. However, WDS systems are capable of more reliable quantitative elemental analysis compared to EDS. A W-filament Cameca SX-100 EPMA was used for this study. EDS detectors are usually fitted with a Be-window to protect the detector from contamination and this results in the absorption of X-rays generated from low atomic number elements[31]. WDS systems do not require these windows, so are able to detect a greater number of low atomic number elements such as oxygen, carbon and nitrogen. The EPMA was operated at a voltage of 15 kV with a beam current of 10 nA, producing a beam-size of ~1 µm at the sample surface. Carbon content can be calculated from the stoichiometric balance after subtracting the recorded heavier elements from a 100 wt% total. However, this technique is only accurate if other elements such as impurities are not present in large quantities.

Spectroscopic analysis was carried out using a LECO CS844 combustion infrared spectrometer, LECO ONH836 inert gas fusion spectrometer and Panalytical Axios wavelength dispersive X-ray fluorescence spectrometer (XRF), which revealed that impurities in the alloy were below the measurable 1000 ppm limit, allowing us to deduce the carbon level in the π-ferrosilicide phase with greater confidence.

**Electron diffraction tomography**. A 10 by 5 µm lamella was extracted from the RR2450 alloy using an FEI Quanta 3D focussed ion beam. We confirmed the thickness of the lamella to be 40 nm through the measurement of interference fringes of convergent beam electron diffraction (CBED) patterns[32] using an FEI Tecnai T20 TEM. The EDT data set was collected using an FEI Tecnai F30 TEM operating at 300 kV. The sample was placed into a Fischione tomography holder, which was used throughout data acquisition. A condenser aperture of 10 µm was used to produce a semi-parallel beam of ~50–100 nm in diameter incident to the sample surface. A Gatan Ultrascan4000 CCD camera captured TEM images and electron diffraction patterns with an image size of 4096×4096 pixels and 16-bit dynamic range. A single crystal of the π-ferrosilicide phase was selected from the lamella, within which the electron beam remains at a stationary point while the crystal is rotated across a wide range (up to −70° to +70°) around the holder axis at 1° increments. In the case of this study we used a minimum tilt-range of −50° to +50° which still provided sufficient data to produce a complete solution. Imaging was carried out using STEM mode between each tilt step, ensuring that the crystal position relative to the incident beam remained constant. Therefore, we could

generate a suitable range of diffraction patterns to reconstruct the 3D reciprocal space using ADT3D according to the method developed by Kolb et al.[14–16]. A benefit of this technique is that the tilt can be conducted around an arbitrary crystallographic axis, which reduces the presence of dynamical artefacts within the generated diffraction patterns[13]. The entire EDT data collection was automated through the use of an automated acquisition module developed for FEI microscopes[15].

In order to produce an optimised data set, we combined electron diffraction with electron beam precession using a NANOmegas P1000 control unit. The precession angle was set to 1.0° deviation from the optic axis at a frequency of 100 Hz. Electron beam precession (developed by Vincent and Midgley[33]) allows us to 'wobble' the Ewald sphere while recording the diffraction patterns, such that the resulting diffracted spot intensities are formed from the integration across the Bragg condition. This has the effect of reducing the dynamical artefacts, which would make the determination of the crystal system inaccurate or impossible to achieve[16]. The precession intensities can be also treated as pseudo-kinematical, whereby access to 3-dimensional atomic structural information becomes more possible[13,16,34]. Diffraction spot indexing and intensity integration for the EDT data set was carried out according to the process described by Mugnaioli et al.[35]. The typical workflow for a structural solution using the EDT technique is shown schematically in Supplementary Fig. 2.

The kinematical approach can be successfully used for determination of the approximate crystal structures, but provides structure models with limited accuracy, because the dynamical diffraction effects inevitably present in the electron diffraction data are not accounted for. To confirm the structure found by the kinematical approach and in particular to verify the details like the potential interstitial carbon position in the structure, dynamical refinement was performed. This method uses the same data collection method as described above, but the model intensities used in the structure refinement process are calculated using full dynamical diffraction theory. The method was described and validated by Palatinus et al.[19,20] and later employed in a number of studies confirming its accuracy and sensitivity to weak features in the electrostatic potential[36,37]. The sample for the data collection on the carbon-containing π-ferrosilicide was prepared in the form of a thin lamella by ion milling a 100 µm thick slice of HIP consolidated RR2450 using the Technoorg Linda ion mill Unimill IV7. The data were collected on the TEM Philips CM120 equipped with the CCD camera Olympus Veleta and the precession unit NANOmegas Digistar. Three data sets were collected. The structure determination provided very similar results and therefore the results obtained on only one data set are presented here. The details of the data collection and refinement are summarised in Supplementary Table 3 and reported atomic positions of the carbon-containing π-ferrosilicide are shown in Supplementary Table 4. The most important result of the dynamical refinement was that the additional maximum, which appeared in the kinematical solution and which seemed to indicate an additional carbon position in the structure, was not confirmed and turned out to be an artefact of the kinematical treatment. All data were processed using the software packages PETS[21] and JANA 2006[22]. In all refinements, the calculated lattice parameters from XRD measurements were used to enhance accuracy.

**Hardness testing**. Nanohardness indentations were made in 10×10 indent grids across microstructurally representative regions in both the RR2450 alloy and carbon-deficient silicide phase ingot. A 45×45 µm nano-indent grid was created in the RR2450 alloy using a Berkovich tip fitted to a Hysitron TI 950 Triboindenter. Indents were made to a 50 nm depth in an attempt to avoid interaction with subsurface features and surrounding grains and precipitates within the microstructures. As the microstructure of the RR2450 alloy is so complex, it was necessary to carry out EBSD over the indent area so that the grain phase could be matched to each indent as demonstrated in Supplementary Fig. 3a. Since the grains were larger in the cast carbon-deficient silicide ingot, a 90×90 µm grid was used instead to ensure that multiple differently orientated grains were sampled. EBSD was also used to determine the phase at each indent position in the cast carbon-deficient ingot sample as shown in Supplementary Fig. 3b. The results of the nanohardness testing of both the RR2450 and silicide ingot alloys, are presented in Fig. 5a.

**Compression testing and neutron diffraction**. In-situ compression testing of the RR2450 alloy was carried out at the Rutherford Laboratory, Didcot, UK, utilising the ENGIN-X time-of-flight (TOF) neutron diffraction beamline[38]. The experimental setup is described schematically in Supplementary Fig. 4. The nature of the ENGIN-X beamline design means that neutron diffraction patterns can be collected in both the axial and transverse directions in the material. Cylindrical samples of 8 mm diameter, 16 mm length, were mounted in the compression rig with a 5 MPa compressive pre-load at which point the first measurement was made. An 8×8 beam footprint was used to measure the samples, at a sampling rate of 25 Hz across a d-spacing range of 0.56 to 2.35 Å. A 30-min counting time at each load step was used. A strain rate of 0.01 %s$^{-1}$ was used to increase compressive stress at nine subsequent 45 MPa intervals, up to 410 MPa which represents the elastic response region of the RR2450 alloy. Stress control was used up to this point in order to ensure constant stresses were applied while measurements were made, since metals will undergo stress relaxation over extended durations.

Ten further measurements were made in the plastic response region of the alloy, at equal intervals, up to a strain (ε) of 1.5%. Strain control was used in this region to maintain a constant strain on the samples to counter the effects of relaxation.

Analysis of the data was carried out using the Generalised Structure Analysis System (GSAS) software package[39] with the EXPGUI[40] front-end package installed. Rietveld refinement of the TOF data was carried out in GSAS allowing the lattice parameters of each phase, at each load-step to be calculated, and is subsequently plotted with respect to the compressive stress at each sampling point.

**Simulated HIP cycle and synchrotron X-ray diffraction**. Simulated HIP cycles were carried out on RR2450 alloy gas-atomised powders. A typical HIP cycle profile is shown in Supplementary Fig. 5. Heat treatments were carried out in situ at the European Synchrotron Radiation Facility (ESRF), Grenoble, France, using beamline ID22. The beamline is fitted with a multi-analyser crystal detector arrangement[41] allowing an achievable resolution of 0.006°. A beam energy of 45 keV (λ = 0.275 Å) was used, ensuring that the effects of x-ray absorption in the material could be minimised.

Powders placed into 1 mm diameter capillaries (0.05 mm wall thickness) were sealed under an average vacuum pressure of 0.035 mbar in order to minimise oxidation. The capillaries were placed on a spinner within a capillary furnace and heated in the ramp-up stage of the HIP cycle from room temperature (25 °C) in 100 °C increments. At each increment point a powder pattern was collected across the d-spacing range 2.63 to 1.05 Å, taking 15 min for collection to complete. The heater would then increase the temperature at a rate of 10 °C min$^{-1}$ until the maximum temperature is obtained. After some dwell time at the maximum cycle temperature, during which, eight 30-min pattern collections were made, the cycle would proceed with the cooling or ramp-down stage, whereby reverse conditions were applied to analyse the samples as they cooled from the dwell temperature back to room temperature.

The patterns were analysed using TOPAS V4.2[30] with peaks fitted using the Pearson VII peak-shape. Achieved goodness of fits were consistently below 2, indicating the quality of the fitted peaks to the experimental pattern.

**Elemental analysis**. The chemical composition of gas-atomised powders of both the RR2450 and Tristelle 5183 alloys was determined through analysis using a LECO CS844 combustion infrared spectrometer, LECO ONH836 inert gas fusion spectrometer and Panalytical Axios wavelength dispersive XRF spectrometer. This study allowed metallic and light element (C, N and O) detection and quantification with suggested errors between 2 to 7%[42]. Phase fraction analysis of the same sets of powders was carried out using a Bruker D8-Discover laboratory XRD diffractometer, fitted with a Co source (λ = 1.79 Å), with subsequent analysis carried out using TOPAS V4.2[30].

**Data availability**. The X-ray crystallographic information files for structures reported in this study have been deposited at the Cambridge Crystallographic Data Centre (CCDC), under deposition numbers 1826958–1826959. These data can be obtained free of charge from The Cambridge Crystallographic Data Centre via http://www.ccdc.cam.ac.uk/data_request/cif.

Files for both the π-ferrosilicide and carbon-deficient silicide phases are also available for download as supplementary data sets. See Data set 1A and 1B for CIF and FCF files of carbon-containing π-ferrosilicide dynamical refinement and Data set 2A and 2B for CIF and FCF files of carbon-deficient π-ferrosilicide dynamical refinement.

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

## Acknowledgements

We thank Rolls-Royce plc. and the Engineering and Physical Sciences Research Council (EPSRC) for financial support provided through the Advanced Metallic Systems Centre for Doctoral Training. M.P. also acknowledges his EPSRC Leadership Fellowship support (EP/I005420/1) and NNUMAN (EP/J021172/1). D.B. and M.P. also acknowledge additional EPSRC funding (EP/R000956/1). We are also grateful to the Stipendienstiftung Rheinland-Pfalz for financial support. Many thanks to Professor D. Dye and Dr. M Rahman from Imperial College London for allowing us access to their casting facilities to produce the carbon-deficient silicide ingot. Thanks to the ID22 staff at the ESRF: Dr. A. Fitch and Dr. C. Giacobbe and the ENGIN-X staff: Dr. S. Kabra and Dr. J. Kelleher for their assistance during set up and data analysis. Finally, we wish to thank the numerous staff and experimental officers at the University of Manchester for their on-going and invaluable assistance with this work, particularly; Prof. M.G. Burke, Dr. J.E. Warren, Dr. J Fellowes, Dr. A. Garner, Mr. G. Harrison, Mr. M. Faulkner, Mr. M. Smith, Mr. A. Forrest and Mr. K. Gyves.

## Author contributions

D.B. carried out the chemical analysis, nanoindentation, compression testing, XRD, EBSD, lamella thinning and CBED thickness measurements and assisted Y.K. with the initial data collection and refinements of the silicide phase structural solution. Y.K., L.P. and S.P.-R. performed EDT measurements, followed by reconstruction of the lattice and space group determination of the silicide phase using kinematical and dynamical refinements. D.T. produced the initial silicide phase lamella and carried out the first EDT tilt-series collection. E.S. calibrated the electron beam precession. U.K., D.S. and M.P. supervised this work and contributed to the discussion.

## Additional information

**Competing interests:** The authors declare no competing interests.

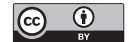

