## [Peer Review File · Nature Communications]

Reviewers' comments:

Reviewer #1 (Remarks to the Author):

Review

Silicide strengthened steel:

The discovery of a new phase and its wider implications

The authors report the discovery of a new silicide phase in steels, and characterize this phase using different techniques and comment on the wider implications of the presence of this phase in the design of novel stainless steels. This work is a serious study, with carefully carried out tests, and relevant discussions. The results are interesting, yet, whether they are as high impact as the authors propose, is not fully clear to the reviewer. (It might be a bit too early to name this phase Bowdenite for 'simplicity'). Here are some comments that may help improve this point:

¥ The authors present an interesting discussion on why this phase would have a large impact in stainless steel design. This may well be the case, however, the work currently only discusses the hardness of the phase, its carbon solubility and structure. Could the authors provide more quantitative mechanical proof on why the steels with these silicide phase (will) surpass others? Since the authors already have 2 produced alloys, one expects to see more property comparisons (tensile properties, high temperature properties, microstructural deformation characteristics etc.), rather than only hardness. The involved groups certainly have a lot of expertise and capabilities in these directions, and perhaps some of these experiments have already been carried out. It would really help to make the implied strong impact more visible if the authors could incorporate more property data.

¥ Following on the same point: silicide phase is softer than carbides, which suggest that a larger fraction of silicide phase would be required to have the same hardness in a multi-phase system. Would there be any other implications of this, property-wise?

¥ While the paper reveals little on these property aspects and focuses more on the phase itself, there seems to be little discussion about the formation mechanism of this phase and its dependence to composition. Why this composition? what can and what cannot be changed? Note that the history of steels with silicide phases are not new. The authors should refer to earlier published work to discuss this, especially those in high impact journals (See Hughes, A new silicide in a 12 per cent Chromium Steel, 1959, Nature).

¥ The EBSD data in Figure 2 reveals that the Silicide phase 'grain size' is larger than that of the other phases and maybe equal to austenite. The nanohardness data, on the other hand, has larger scatter than even those of the carbides. The SEM-EDX data suggests a homogeneous composition, and the author state that this is confirmed with WDS. Why is the scatter then? Are the authors sure this is a homogeneous phase? Figure S5 suggests that there are few indents that hit the bulk of the phase directly, and many indents are near boundaries. If this is the case, the hardness data must be influenced from the boundary effect. In other words, the phase is actually softer than suggested in the paper.

¥ Also a non-technical point: By submitting to Nature Communications, the authors show their will to address a wide audience, but in absence of some of the materials processing details, it would be impossible to repeat these experiments. Although the reviewer understands the IP challenges of companies such as Sandvik and RR in sharing some of the details of alloy production, past alloy-design work published in high impact journals (see, e.g., gum metal literature) did create an 'artificial' impact, due to similar "hidden" issues. Tough situation for the authors of course, but worth further consideration...

Reviewer #2 (Remarks to the Author):

The manuscript reports the discovery of a new phase in the alloy RR2450, describes the structure of the phase and reports on its exceptionally high hardness. Based on the context provided in the manuscript the findings appear very interesting, but I myself, not being an expert in metallurgy, cannot judge authoritatively on the uniqueness of ground-breaking nature of the discovery.

One of the main aspects of the work is the crystallographic analysis of the newly discovered phase. The analysis is performed using the ADT technique, which is known to often provide very reliable structure models from very small crystals. However, it is also known to provide results which may not be perfectly accurate and which need independent validation. This is especially true for the detection of very weak signals in the Coulomb potential.

I have studied the reported results very carefully and my opinion is that the presented results do not provide sufficient evidence for the presence of the carbon atom in the place indicated by the authors and for the lowering of the symmetry from the prototype space group P213 to R3. When reading the manuscript I have not found a convincing answer to the following questions and problems:

- What is the height of the potential peak assigned to the carbon atom compared to the noise in the potential map? In Fig. 3a it seems that there are many noise peaks of the same height as the supposed carbon peak. In Fig 3b the "carbon" potential is missing, but there are several peaks of the same height nearby. Thus, are the "carbon" peaks significant or are they at the same level as the noise?
- What are the arguments for lowering the symmetry from P213 to R3? From the text it seems that the only "hard" argument is that when solved in P1, the solution in Sir2014 placed the carbon atom in only one out of four equivalent sites in P213. However, this alone cannot be taken as a proof, but merely an indication. If the additional peak is not a genuine carbon atom, but just a noise peak, it may well appear only in one place due to the uneven distribution of the noise in the data.
- How were the atomic types of the heavier atoms assigned to the maxima in the potential? There are Fe, Ni and Cr atoms present, all of which have so close scattering factors that they are impossible to distinguish by the approximate kinematical analysis of the electron diffraction data. It appears that the atomic types were left as they were assigned by SIR2014, which would mean they were assigned essentially randomly. While the carbon-containing pi-ferrosilicide has Fe, Cr and Ni atomic types assigned, the carbon-deficient structure is presented with only iron in the structure model. Why?
- The violation of the systematic absences of the P213 space group is presented as an indication of the symmetry lowering from P213 to R3. However, this violation can occur also due to multiple scattering, and in Fig. S3 the geometry of the diffraction pattern is such that multiple diffraction effects are indeed likely in that particular crystal orientation. Have any more detailed tests been performed to confirm that the violations are genuine and not only due to multiple/dynamical scattering effects? Without additional tests the observation in Fig.S3 cannot be used as a documentation of the symmetry lowering.
- Probably the strongest indication that the potential at the carbon position is genuine is provided by the difference potential map, which shows the potential at the carbon position. It is, however, not clear from the image, if the potential map is shown in the full unit cell or if only a part of the potential around the carbon position is shown. It seems somewhat surprising that such a weak signal would be so clearly above all the noise in the difference map. It would be more appropriate to set the isosurface level so that also the highest noise peaks in the difference map are visible and the significance of the difference potential at the supposed carbon position can be judged.
- In fig. 3c the difference potential has a shape of a dumbbell, yet only one atom (in one lobe of the dumbbell) is placed in the unit cell. Why?
- In the text it is stated that the atom occupies a trigonal bipyramidal site. However, inspection of the provided CIF shows that the carbon atom is located on an almost perfect tetrahedral

coordination.

- The interatomic distances are not discussed in the text. Have the distances of the carbon atom to its neighbors been considered? The distances measured in the deposited CIF are the following: $d(\text{C-Si})=1.51\text{\AA}$, $d(\text{C-Fe})=1.57\text{\AA}$. Both these distances are extremely short for such contacts, the typical C-Si distance being $>1.8\text{\AA}$ and C-Fe distance $>1.7\text{\AA}$. Do the authors have an explanation why the interatomic distances are so short? To me this is a strong indication that the atomic position is not real, because there is not enough place for a carbon atom in that void.

To conclude, I see many problems and open questions connected to the claim that the real structure of carbon-rich pi-ferrosilicide really contains a carbon site at the suggested place, and that the symmetry is lower than P213. In my opinion the only reliable way of proving the claims would be to perform a detailed structure refinement and show that the model with the carbon atom gives a significantly better fit to the data than the model without it, and that the symmetry lowering also improves the fit. Unfortunately, the refinement using kinematical approximation is most likely not sensitive enough to provide such distinction. The refinement of electron diffraction data using the dynamical diffraction theory, which has become available recently, may provide a much more accurate result.

An alternative option would be to collect several data sets, process them the same way as described in the manuscript, and show that the additional potential and the symmetry lowering can be observed independently in all (or at least most) data sets. Such reproducibility would provide a strong support that the feature is genuine.

If this problem is solved and the manuscript is modified accordingly, I will be happy to review the manuscript once more and provide more detailed comments on the text.

Actions/responses addressing reviewers' comments

The authors would like to thank the referees for their comments and suggestions regarding our manuscript entitled: "Silicide strengthened steel: The discovery of a new phase and its wider implications". Below, we present a description of the work included in the manuscript which now addresses these comments. These extra elements of analysis have involved conducting a dynamical refinement of the dataset to establish whether the presence of a carbon atomic site was in fact genuine as previously reported.

In the course of these revisions, we have significantly advanced our understanding of the π -ferrosilicide crystallography based on the reviewers' previous feedback. By utilising the latest dynamical refinement techniques with our electron diffraction tomography dataset, we have been able to show that a discrete carbon site does not exist within this phase, but in fact that carbon must be distributed randomly at existing atomic sites. This is supported by data now added to the manuscript demonstrating the decomposition of the π -ferrosilicide into δ -ferrite and M_7C_3 carbides during a simulated hot isostatic pressing cycle, carried out in-situ with synchrotron x-ray diffraction. We believe that this work adds significant value not only in the field of materials science but also within the field of crystallography, since we are able to highlight the possible pitfalls of the kinematical technique, and the kinds of artefacts this kind of refinement can produce.

Reviewer #1 (Remarks to the Author):

¥ The authors present an interesting discussion on why this phase would have a large impact in stainless steel design. This may well be the case, however, the work currently only discusses the hardness of the phase, its carbon solubility and structure. Could the authors provide more quantitative mechanical proof on why the steels with these silicide phase (will) surpass others? Since the authors already have 2 produced alloys, one expects to see more property comparisons (tensile properties, high temperature properties, microstructural deformation characteristics etc.), rather than only hardness. The involved groups certainly have a lot of expertise and capabilities in these directions, and perhaps some of these experiments have already been carried out. It would really help to make the implied strong impact more visible if the authors could incorporate more property data.

We have included results from an in-situ compression loading neutron diffraction study to further enhance the point of the high strength of the silicide phase. The results show how austenite and ferrite yield at 410 and 853 MPa respectively, whilst the load is partitioned to the π -ferrosilicide phase, which does not yield or undergo brittle fracture, despite loading in excess of 1.6 GPa (see Figure 5b).

¥ Following on the same point: silicide phase is softer than carbides, which suggest that a larger fraction of silicide phase would be required to have the same hardness in a multi-phase system. Would there be any other implications of this, property-wise?

The evolution of the π -ferrosilicide phase during a simulated HIP cycle is now explored in the text (see Figure 6). We have shown that the π -ferrosilicide precipitates from the δ -ferrite during the cooling phase of the HIP cycle, which will lead to a more even distribution of hard phase, as opposed to carbides (particularly NbC) which forms in the melt before gas-atomisation of the powder is carried out, leading to carbide clusters in the final HIPed product. In addition, we have shown that an orientation relationship exists

between the π -ferrosilicide phase and the δ -ferrite, described by: $\{100\}_\delta \parallel \{110\}_\pi$, $\langle 010 \rangle_\delta \parallel \langle \bar{1}13 \rangle_\pi$ and $\{100\}_\delta \parallel \{120\}_\pi$, $\langle 010 \rangle_\delta \parallel \langle \bar{2}10 \rangle_\pi$. Since coherency between the π -ferrosilicide and δ -ferrite is improved, these alloys are less likely to suffer from grain pull-out in-service, unlike carbides which often do not form coherently with the parent matrix.

¥ While the paper reveals little on these property aspects and focuses more on the phase itself, there seems to be little discussion about the formation mechanism of this phase and its dependence to composition. Why this composition? what can and what cannot be changed? Note that the history of steels with silicide phases are not new. The authors should refer to earlier published work to discuss this, especially those in high impact journals (See Hughes, A new silicide in a 12 per cent Chromium Steel, 1959, Nature).

The evolution of the π -ferrosilicide phase during a typical HIP cycle was explored using in situ synchrotron x-ray diffraction analysis. We can show how the silicide decomposes and precipitates at a temperature of 920 °C (see Figure 6). In addition, we have included data demonstrating how the variation in alloy chemistry has a very clear effect of the π -ferrosilicide fraction evolved (Figure 7). In particular, we isolate the balance of Ni/Si as the primary elements required to control the triplex matrix balance between one that is austenitic versus one that is predominantly π -ferrosilicide/ferrite.

¥ The EBSD data in Figure 2 reveals that the Silicide phase 'grain size' is larger than that of the other phases and maybe equal to austenite. The nanohardness data, on the other hand, has larger scatter than even those of the carbides. The SEM-EDX data suggests a homogeneous composition, and the author state that this is confirmed with WDS. Why is the scatter then? Are the authors sure this is a homogeneous phase? Figure S5 suggests that there are few indents that hit the bulk of the phase directly, and many indents are near boundaries. If this is the case, the hardness data must be influenced from the boundary effect. In other words, the phase is actually softer than suggested in the paper.

Indents made near the boundary were excluded from the data used to calculate the average hardness. The scatter within the dataset is likely the result of sampling a range of orientations producing differences in observed hardness, as well as possible sub-surface features such as carbides which are inadvertently sampled. These issues are now clearly stated in the text.

¥ Also a non-technical point: By submitting to Nature Communications, the authors show their will to address a wide audience, but in absence of some of the materials processing details, it would be impossible to repeat these experiments. Although the reviewer understands the IP challenges of companies such as Sandvik and RR in sharing some of the details of alloy production, past alloy-design work published in high impact journals (see, e.g., gum metal literature) did create an 'artificial' impact, due to similar "hidden" issues. Tough situation for the authors of course, but worth further consideration...

It is possible to cast an ingot of the π -ferrosilicide based purely on the compositional data supplied in the data. Specific conditions within the vacuum arc melter are not required to replicate this work. Unfortunately, RR IP would not allow inclusion of specific HIP cycle parameters but generally manufacturers such as Sandvik will offer standard cycles for steels which will produce comparable microstructures to those in this study.

Reviewer #2 (Remarks to the Author):

- What is the height of the potential peak assigned to the carbon atom compared to the noise in the potential map? In Fig. 3a it seems that there are many noise peaks of the same height as the supposed carbon peak. In Fig 3b the "carbon" potential is missing, but there are several peaks of the same height nearby. Thus, are the "carbon" peaks significant or are they at the same level as the noise?

Sigma values are now indicated on these maps which have been plotted for both the kinematical and dynamical cases, although this point is now largely redundant as we have shown that the peaks were indeed artefacts from the kinematical refinement.

- What are the arguments for lowering the symmetry from P213 to R3? From the text it seems that the only "hard" argument is that when solved in P1, the solution in Sir2014 placed the carbon atom in only one out of four equivalent sites in P213. However, this alone cannot be taken as a proof, but merely an indication. If the additional peak is not a genuine carbon atom, but just a noise peak, it may well appear only in one place due to the uneven distribution of the noise in the data.

Again, this point is now redundant as we have shown that a specific carbon site cannot be located using a more accurate dynamical refinement of the dataset. Therefore, symmetry for the π -ferrosilicide remains as P2₁3.

- How were the atomic types of the heavier atoms assigned to the maxima in the potential? There are Fe, Ni and Cr atoms present, all of which have so close scattering factors that they are impossible to distinguish by the approximate kinematical analysis of the electron diffraction data. It appears that the atomic types were left as they were assigned by SIR2014, which would mean they were assigned essentially randomly. While the carbon-containing π -ferrosilicide has Fe, Cr and Ni atomic types assigned, the carbon-deficient structure is presented with only iron in the structure model. Why?

This was not made clear previously and has been corrected in the text, but we assume that the same sites can be occupied by Fe, Ni and Cr due to the similar scattering factors of these elements. This is now clear in Figure 4 where the schematic of the π -ferrosilicide unit cell which states that the red atomic sites are interchangeable.

- The violation of the systematic absences of the P213 space group is presented as an indication of the symmetry lowering from P213 to R3. However, this violation can occur also due to multiple scattering, and in Fig. S3 the geometry of the diffraction pattern is such that multiple diffraction effects are indeed likely in that particular crystal orientation. Have any more detailed tests been performed to confirm that the violations are genuine and not only due to multiple/dynamical scattering effects? Without additional tests the observation in Fig.S3 cannot be used as a documentation of the symmetry lowering.

Dynamical refinement of the data has now been carried out, and this has shown that multiple scattering has influenced the kinematic refinement and suggested that a unit cell distortion would reduce the symmetry to R3. We have now shown that this distortion is actually not present and the original space group P2₁3 is retained.

- Probably the strongest indication that the potential at the carbon position is genuine is

provided by the difference potential map, which shows the potential at the carbon position. It is, however, not clear from the image, if the potential map is shown in the full unit cell or if only a part of the potential around the carbon position is shown. It seems somewhat surprising that such a weak signal would be so clearly above all the noise in the difference map. It would be more appropriate to set the isosurface level so that also the highest noise peaks in the difference map are visible and the significance of the difference potential at the supposed carbon position can be judged.

This point is now redundant as the figure has been removed based on the revised results from the dynamical refinement.

- In fig. 3c the difference potential has a shape of a dumbbell, yet only one atom (in one lobe of the dumbbell) is placed in the unit cell. Why?

This point is now redundant as the figure has been removed based on the revised results from the dynamical refinement.

- In the text it is stated that the atom occupies a trigonal bipyramidal site. However, inspection of the provided CIF shows that the carbon atom is located on an almost perfect tetrahedral coordination.

This point is now redundant as the carbon atom is shown not to be present, based on the revised results from the dynamical refinement.

- The interatomic distances are not discussed in the text. Have the distances of the carbon atom to its neighbors been considered? The distances measured in the deposited CIF are the following: $d(\text{C-Si})=1.51\text{\AA}$, $d(\text{C-Fe})=1.57\text{\AA}$. Both these distances are extremely short for such contacts, the typical C-Si distance being $>1.8\text{\AA}$ and C-Fe distance $>1.7\text{\AA}$. Do the authors have an explanation why the interatomic distances are so short? To me this is a strong indication that the atomic position is not real, because there is not enough place for a carbon atom in that void.

This point is now redundant as the carbon atom is shown not to be present, based on the revised results from the dynamical refinement.

Reviewers' comments:

Reviewer #1 (Remarks to the Author):

Thanks for the updated manuscript with the modifications.

Reviewer #3 (Remarks to the Author):

The crystal structure determination of pi-ferrosilicides is thoroughly done. The space symmetry, unit cell and initial structure model have been elucidated from kinematical electron diffraction tomography data, whereas the final atomic positions and occupancy factors were determined from a dynamical refinement. However, the presentation of the refinement results must be improved significantly before the manuscript could be accepted for publication.

1) Unit cell parameters from Table S3 are different from those in the CIF files. Provide the same set of the unit cell parameters (as determined from EDT or PXRD) with their standard deviations.

2) Is ~22% difference in density between the ferrosilicides with and without carbon real? Please, check.

3) There is something wrong with the crystal dimensions of the pi-ferrosilicide with carbon (125663 nm²?). Please, check.

4) Explain why so many refined parameters for such simple structures.

5) Atomic coordinates in Table S4 differ from those in the CIF file 1A. Please, make them consistent.

6) Provide standard deviations for the refined atomic coordinates in Table S4 and the CIF files.

7) As the crystal structure was refined in anisotropic approximation for the ADPs, Uiso should be replaced by Ueq in Table S4. Provide the expression for calculation of Ueq from the ADP tensor components. Provide the units for Ueq.

8) Provide list of interatomic distances for every refined structure.

9) Verify the numbers of the collected frames in Table S4 (100 or 121) and in Fig.S2 (141). Why do they differ?

Discussion on possible C position must be shortened as it appears to be an artifact. There is no sense to mark this position in Fig. 3a. There is no sense to provide the results of the kinematical refinement as the dynamical refinement are clearly superior.

Finally, it seems that the title "Silicide strengthened steel: The discovery of a new phase and its wider implications" does not reflect the actual content of the article. It does not give a clue on the nature of the new phase. "Wider implications" sounds too vague and uncertain.

Actions/responses addressing reviewers' comments

The authors would like to thank the referees for their comments and suggestions regarding our manuscript entitled: "Silicide strengthened steel: The discovery of a new phase and its potential for use in wear-resistant applications". Below, we present a description of the corrections included in the manuscript which now addresses these comments.

Reviewer #3 (Remarks to the Author):

The crystal structure determination of pi-ferrosilicides is thoroughly done. The space symmetry, unit cell and initial structure model have been elucidated from kinematical electron diffraction tomography data, whereas the final atomic positions and occupancy factors were determined from a dynamical refinement. However, the presentation of the refinement results must be improved significantly before the manuscript could be accepted for publication.

We appreciate the thorough check made on the crystallographic data by the referee. We discovered that part of the problem arose from a mistake on our side. We uploaded a non-final version of the CIF files. Now we have rerun the refinements once more with lattice parameters set consistently to the lattice parameters from powder x-ray diffraction, cross-checked the applied parameters again for their mutual consistency, and uploaded the consistent CIF files.

1) Unit cell parameters from Table S3 are different from those in the CIF files. Provide the same set of the unit cell parameters (as determined from EDT or PXRD) with their standard deviations.

Done. All unit cell parameters reported in the manuscript and in the CIFs are now based on the powder XRD data, as they are more accurate than the electron diffraction data.

2) Is ~22% difference in density between the ferrosilicides with and without carbon real? Please, check.

This was a typo in the table. Corrected.

3) There is something wrong with the crystal dimensions of the pi-ferrosilicide with carbon (125663 nm²?). Please, check.

This number corresponds to the illuminated area of the crystal, and it is calculated from the beam diameter of 400 nm. The number was rounded to whole thousands and an explanatory note was added to the table to make this clear.

4) Explain why so many refined parameters for such simple structures.

A large number of parameters are scale factors of individual frames. This is a standard procedure in the dynamical refinement. For the sake of clarity, the parameters were split

to two parts in table S3 – structural and scale parameters, and an explanatory note was added to the table.

5) Atomic coordinates in Table S4 differ from those in the CIF file 1A. Please, make them consistent.

Corrected.

6) Provide standard deviations for the refined atomic coordinates in Table S4 and the CIF files.

Corrected

7) As the crystal structure was refined in anisotropic approximation for the ADPs, Uiso should be replaced by Ueq in Table S4. Provide the expression for calculation of Ueq from the ADP tensor components. Provide the units for Ueq.

Corrected

8) Provide list of interatomic distances for every refined structure.

The list was added in the CIF files and also as new supplementary tables, S5a and S5b for both carbon-containing and carbon-deficient dynamically refined structures respectively.

9) Verify the numbers of the collected frames in Table S4 (100 or 121) and in Fig.S2 (141). Why do they differ?

Each data collection had a different number of frames due to experimental limitations (tilt range, sample holder shading). Fig S2 was modified to reflect this possible ambiguity and it is made clearer in the Methods section that we used a reduced tilt-range to produce a complete solution.

Discussion on possible C position must be shortened as it appears to be an artifact. There is no sense to mark this position in Fig. 3a. There is no sense to provide the results of the kinematical refinement as the dynamical refinement are clearly superior.

We have shortened the discussion around the carbon position, particularly in relation to our earlier refinements using P1 symmetry as these aspects no longer add to the paper. However, we do feel that retaining data for comparison between the kinematical and dynamical datasets is important, as it serves to highlight the limitations and potential pitfalls of relying purely on the kinematical technique and clearly demonstrates the superiority of the dynamical approach in this application. We have made this reason for retaining this in the main text clearer to the reader also.

Finally, it seems that the title “Silicide strengthened steel: The discovery of a new phase and its wider implications” does not reflect the actual content of the article. It does not give a clue on the nature of the new phase. “Wider implications” sounds too vague and uncertain.

The title of the paper has been amended to remove ambiguity in “wider implications”. The new title reflects the original intent of this work, in developing high-strength steels for wear-resistant applications.

REVIEWERS' COMMENTS:

Reviewer #3 (Remarks to the Author):

The authors satisfactorily replied to all comments and introduced appropriate corrections. The manuscript can be accepted as it is.